# Maintenance Process Simulation Based Maintainability Evaluation by Using Stochastic Colored Petri Net

**Zhong Lu [1],*, Jie Liu [2], Li Dong [1] and Xihui Liang [3]**

1   College of Civil Aviation, Nanjing University of Aeronautics and Astronautics, Nanjing 211106, China
2   School of Transportation Science and Engineering, Beihang University, Beijing 100191, China
3   Department of Mechanical Engineering, University of Manitoba, Winnipeg, MB R3T5V6, Canada
*   Correspondence: luzhong@nuaa.edu.cn

**Abstract:** Maintainability is a critical design characteristic of products. Maintainability evaluation plays an important role in maintainability design. Existing maintainability evaluation approaches neglect logistic effects on system maintainability. In this paper, tuples of stochastic colored petri nets (SCPN) are used to express the constituents of a maintenance process; and the maintenance process model is developed based on the SCPN. Approaches for calculating the required maintenance resources are proposed according to the state equation of the SCPN, and a method for calculating maintenance time is proposed by using an SCPN based discrete-event simulation. Finally, the line maintenance of a wheel steering system is used as a case study to illustrate the application and effectiveness of our proposed approaches. The work discussed herein provides a maintainability evaluation methodology based on the maintenance task demonstration that is conducted on the digital mockup of products. The approaches can be applied in the design stage when there are no physical mockups, and the maintainability design can be carried on concurrently with the development of products.

**Keywords:** maintainability modeling; maintainability evaluation; logistics support analysis; stochastic colored petri net; discrete-event simulation

## 1. Introduction

Maintainability, a characteristic of design, affected by various personnel and logistic factors, is one of the significant requirements that must be considered during product development. Maintainability evaluation is an important task specified in MIL-STD-470B "Maintainability Program Requirements for Systems and Equipment". Its objective is to determine, at all levels of maintenance, the impact of the operation, maintenance and support environment on the maintainability parameters.

At present, a maintainability evaluation is still conducted in terms of MIL-STD-471A "Maintainability Verification/Demonstration/Evaluation," which was issued more than 40 years ago. In this standard, the evaluation is mainly focused on time parameters, for instance, the mean time to repair (MTTR). Time samples are obtained by demonstrating the maintenance process on physical or virtual prototypes, and the time parameters can be calculated by statistical methods. Yu et al. [1] have improved the maintainability evaluation given in MIL-STD-471A. A time-based maintainability evaluation approach has been proposed by using the Generalized Reliability Analysis Simulation Program (GRASP). In their approach, a demonstration is not required to be conducted for the whole maintenance process. The time samples of the whole maintenance process can be obtained by simulations in terms of the time distribution of each maintenance task. Barabadi et al. [2] developed the

Cox regression model and its extension in the presence of time-dependent covariates for determining maintainability, and the effect of the operational environment on maintainability is taken into account in the model. Zhou et al. [3] thought that time parameters of maintainability cannot be totally controlled by the corresponding design apartment, and they proposed a new time characteristics-based maintainability allocation method to solve this problem. Guo Z.Y. et al. [4] developed an immersive maintainability verification and evaluation system to improve maintainability in the early design stages. An aero-engine project is presented as a case to demonstrate the effectiveness and feasibility of the system. Peng et al. [5] designed a desktop virtual reality-based system for maintainability design and verification, a case-based reasoning method combined with the extensible markup language are applied in the maintainability verification of the system. Yu et al. [6] also applied a fuzzy comprehensive evaluation in the maintainability evaluation based on the virtual environment, and the methods can be applied along with a maintenance task virtual demonstration. Zhou et al. [7] also studied the effects of ergonomics and human factors in maintenance space evaluation, and human factors are considered as an attribute in maintainability. Wani et al. [8,9] studied the maintainability assessment of mechanical products in design phases based on maintainability factors and attributes, and the maintainability index is assessed by using a digraph and matrix method. Meier and Russell [10] presented a model process for conducting maintainability practices in the construction industry, the benefits of various maintainability practices are illustrated in their work. Chen and Cai [11] studied the effects on system maintainability caused by factors of physical design, logistics support and ergonomics, and a maintainability assessment approach is presented based on the Vector Projection Method. Tjiparuro et al. [12] studied the elements and attributes that will affect system maintainability according to previous research efforts, and a maintainability analysis approach is proposed in terms of functional design and maintainability axioms for the conceptual design. Slavila et al. [13] applied the fuzzy theory to study the maintainability evaluation, and design variables are expressed by linguistic variables in their methods. Desai and Mital [14] addressed the effect of human factors in maintainability design and presented a comprehensive design to improve the maintainability of products. Luo et al. [15] proposed a method based on fuzzy grey relational analysis to determine the prioritization of maintainability indices, and the fuzziness and uncertainty of design factors are considered in their method. Ertas et al. [16] outlined a diagnostic approach to quantify the maintainability of a commercial-off-the-shelf-based system, and their method is given by analyzing the complexity of the deployment of the system components. Tu [17] presented a virtual maintenance application for the maintainability analysis and evaluation of flexible cables, and recommendations on the maintainability of products containing cables can be obtained during the early stages of product design. A case study on aircraft system disassembly was used to validate the feasibility of the improvement for cable maintainability designs. Jian et al. [18] developed an evaluation index system of product maintainability based on the theory of product life cycle. A maintainability calculation method was proposed with consideration of the inherent attributes and external factors simultaneously. Guo C.H. et al. [19] proposed a design approach considering the relationship between maintainability and functional construction. A practical case was given by implementing the proposed approach for the lubrication system of an armored vehicle to validate the effectiveness and feasibility of the approach. The present maintainability evaluation methods can be divided into three types. The first is the time related methods based on the probability theory [1–3], the maintenance time is the only index considered in these methods. The second is the process based method [4–7], the maintenance process is modeled to support task demonstration in these methods. The last is the multiple attribute decision based method [8–19], different kinds of attributes are considered and maintainability deficiencies can be discovered with these methods.

In all the above mentioned methods, logistic factors including the required tools or equipment, spare parts and maintenance personnel are seldom considered in the maintainability evaluation. The maintainability parameters can be divided into two types essentially. The first one is the time related parameters and the second one is the economic parameters. The first type of parameters includes MTTR and repair rate; and the second types of parameters includes direct maintenance

man-hours per hour and direct maintenance cost per hour. The consumption of spare parts, the depreciation cost of tools or equipment, and the labor cost of maintenance personnel are closely related to the economic parameters. The requirement of system availability can be satisfied by controlling the time related parameters, and the economic efficiency of the system is realized by controlling the economic parameters. Therefore, both time and logistic factors should be considered in maintainability evaluations.

As a graphical tool for discrete-event simulation, Petri nets have been widely used in process modeling, the specific application includes reliability and maintainability optimization [20–22], life-cycle cost analysis [23–25], production process modeling [26], and so on. Logistic factors are not considered in these studies either. The stochastic colored Petri net (SCPN) is extended from the classical Petri net by introducing colored tokens and stochastic timed transitions [27]. Compared with the classical Petri nets, the SCPN has two advantages in the modeling of the maintenance process simulation. Firstly, colored tokens can be applied to denote different kinds of maintenance resources as well as system states; secondly, timed transitions can be applied to denote maintenance tasks with time attributes. In this study, a maintenance process model is developed by using the SCPN. Approaches for calculating the required maintenance resources as well as maintenance time are proposed by using the maintenance process model. In this way, both time and logistic factors are considered in our maintainability evaluation approach. The rest of this paper is structured as follows. In Section 2, some basic definitions related to the SCPN are given. In Section 3, tuples of an SCPN are used to express constituents of a maintenance process, and the maintenance process model is developed based on the SCPN. In Section 4, approaches for calculating the required maintenance resources are proposed according to the state equation of the SCPN, and a method for calculating maintenance time is proposed based on discrete-event simulation. In Section 5, a case study using the line maintenance of a wheel steering system is given to illustrate the application of our approaches. In Section 6, concluding remarks are presented.

## 2. Definitions Related to Stochastic Colored Petri Nets

### 2.1. Linear Function L(X)

Let $X = \{x_1, x_2, \cdots, x_k\}$, $L(X)$ is a linear function defined in X with nonnegative integer coefficients. When not all coefficients of $L(X)$ are equal to 0, $L(X)$ will be denoted as $L(X)_+$. We have

$$
\begin{aligned}
L(X) &= a_1 x_1 + a_2 x_2 + \cdots a_k x_k \\
L(X)_+ &= a_1^+ x_1 + a_2^+ x_2 + \cdots a_k^+ x_k
\end{aligned}
\tag{1}
$$

where $a_i$ and $a_i^+$ $(i = 1, 2, \cdots k)$ are both nonnegative integers, and $a_1^+ + a_2^+ + \cdots a_k^+ \neq 0$. Let $L_a(X) = a_1 x_1 + a_2 x_2 + \cdots a_k x_k$ and $L_b(X) = b_1 x_1 + b_2 x_2 + \cdots b_k x_k$, if $\forall i$ $(i = 1, 2, \cdots k)$: $a_i \geq b_i$, we have $L_a(X) \geq L_b(X)$.

### 2.2. Stochastic Colored Petri Net

An SCPN is an 8-tuple $\sum = (S, T, F, C, K, W, M, I)$ such that [27]

i.   $S = \{s_1, s_2, \cdots, s_n\}$ is the set of places.
ii.  $T = \{t_1, t_2, \cdots, t_m\}$ is the set of transitions.
iii. $F \subseteq (S \times T) \cup (T \times S)$ is the set of arcs.
iv.  $C = \{c_1, c_2, \cdots, c_l\}$ is the set of token colors.
v.   $K : S \to L(C)_+$ is the capacity function of the place.
vi.  $W : F \to L(C)_+$ is the weight function of the arc.
vii. $(M : S \to L(C)$ is the marking of the place, and $\forall s \in S : M(s) \leq K(s)$.
viii. $I = \{\eta_1, \eta_2, \cdots, \eta_m\}$ is the set of firing rates, and $\eta_i$ is the firing rate of the $i$th transition.

*2.3. Pre-Set and Post-Set*

∀ $x \in S \cup T$, •$x$ is called the pre-set of $x$ and $x$• is called the post-set of x such that

$$\begin{aligned} \bullet x &= \{y \,|\, (y \in S \cup T) \cap ((y,x) \in F)\} \\ x\bullet &= \{y \,|\, (y \in S \cup T) \cap ((x,y) \in F)\} \end{aligned} \tag{2}$$

*2.4. Enabled and Fired*

A transition $t \in T$ is enabled if and only if [27]

$$\begin{cases} \forall s \in \bullet t : M(s) \geq W(s,t) \\ \forall s \in t\bullet - \bullet t : M(s) + W(t,s) \leq K(s) \\ \forall s \in t\bullet \cap \bullet t : M(s) + W(t,s) - W(s,t) \leq K(s) \end{cases} \tag{3}$$

After the transition $t$ has been fired, the markings of the net will be changed according to the following rule [27]

$$\forall s \in S : \ M'(s) = \begin{cases} M(s) - W(s,t), s \in \bullet t - t\bullet \\ M(s) + W(s,t), s \in t\bullet - \bullet t \\ M(s) + W(t,s) - W(s,t), s \in \bullet t \cap t\bullet \\ M(s), otherwise \end{cases} \tag{4}$$

## 3. Maintenance Process Modeling Based on SCPN

The maintainability of a system can be reflected by all its maintenance actions. In this study, the maintenance actions are divided into three levels, which are maintenance processes, maintenance events and maintenance tasks from top to bottom [28]. The maintenance process includes all possible maintenance events of the system, the maintenance event includes all maintenance tasks pertinent to the replacement or restoration of a replaceable unit, and the maintenance task is the basic element that is not required to be subdivided any more. The hierarchical chart of maintenance actions is given in Figure 1. We can see that the top level maintenance process is composed of $n$ maintenance events, and the maintenance event pertinent to unit $i$ contains $m$ maintenance tasks.

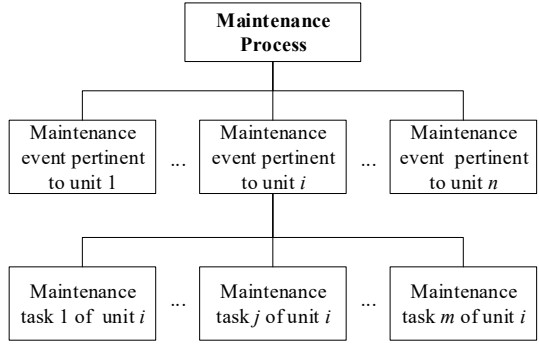

**Figure 1.** Hierarchical chart of maintenance actions.

In the following paragraphs, the tuples of an SCPN are used to express constituents of the maintenance process including maintenance tasks, system states, and maintenance resources such as tools, equipment, spare parts and maintenance personnel, then the maintenance process model is developed based on the SCPN.

*3.1. Maintenance Task Expression*

In Petri nets, transitions are used to describe events that change the state of the system. In maintenance processes, system states will be changed when a maintenance task has been completed.

Hence, timed transitions drawn as hollow bars are used to express maintenance tasks in the maintenance process model. As the time duration of maintenance tasks may follow various distributions, the firing rate of the timed transition may not always be a constant value. We will use the firing time set instead of the firing rate set in the SCPN when modeling the maintenance process. The set of firing time is expressed as $\Pi = \{\pi_1, \pi_2, \cdots, \pi_m\}$, $\pi_i(i = 1, 2, \cdots m)$ is the firing time of the *i*th transition.

In addition to timed transitions, there are also immediate transitions drawn as solid bars in our maintenance process. The immediate transitions have no time attributes, and they will be fired as soon as they are enabled. Namely, the firing time of the immediate transition is 0. In this study, the immediate transitions are usually used in competitive structures, which will be discussed in Section 3.4.3. When there are immediate transitions, it means only one of the several maintenance events can be started.

### 3.2. Maintenance State and Resource Expression

In maintenance processes, the places and their tokens are used to express the system maintenance states and maintenance resources. The maintenance resources are divided into two categories, which are consumable resources and reusable resources. Spare parts are consumable resources, as they will be consumed during the pertinent maintenance tasks. And maintenance personnel, tools and equipment are reusable resources as they will not be consumed during the pertinent maintenance task.

#### 3.2.1. Expression of System States

The system states are expressed by the state places. When there is a token in the state place, it indicates that all the transitions belonging to the pre-set of the state place have just been fired. Namely, all the maintenance tasks expressed by the transitions belonging to the pre-set have just been completed. Thus, the state places only have one kind of token color.

The elements in the pre-set of state places are transitions expressing the preceding maintenance tasks of the states, and the elements in their post-set are transitions expressing the subsequent maintenance tasks of the states. Therefore, the intersection of the pre-set and the post-set of a state places is empty.

Let $S_S = \{s_{s1}, s_{s2}, \cdots, s_{sn_s}\}$ be the set of state places, $n_s$ is the number of state places, and the marking and capacity function of the state place $s_s$ can be expressed as

$$\begin{aligned} M(s_s) &= m_s c_s \\ K(s_s) &= c_s \end{aligned} \tag{5}$$

where $c_s$ is the token color of the state place $s_s$, $m_s$ is the number of token color $c_s$ in the state place $s_s$, it can be either 0 or 1.

#### 3.2.2. Expression of Consumable Resources

Consumable resources are expressed by consumable-resource places. As there are many types of consumable resources in the maintenance process, the model will be very complex if each consumable resource is expressed by an individual consumable-resource place. Therefore, token colors are introduced to express different kinds of consumable resources. In this way, only one consumable-resource place is needed in the maintenance model.

The elements in the pre-set of consumable-resource places are transitions expressing the maintenance tasks of acquiring the corresponding consumable resources, and the elements in their post-set are transitions expressing maintenance tasks of consuming the corresponding consumable resources. Therefore, the intersection of the pre-set and the post-set of consumable-resource places is also empty.

Let $S_c = \{s_c\}$ be the set of consumable-resource places, their marking and capacity function can be expressed as

$$M(s_c) = m_{c1}c_{c1} + m_{c2}c_{c2} + \cdots + m_{cn_c}c_{cn_c}$$
$$K(s_c) = k_{c1}c_{c1} + k_{c2}c_{c2} + \cdots + k_{cn_c}c_{cn_c} \tag{6}$$

where $c_{ci}$ $(i = 1, 2, \cdots, n_c)$ is the $i$th token color expressing the $i$th type of consumable resources; $m_{ci}$ and $k_{ci}$ are both nonnegative integers, the former represents the number of the $i$th token color in the place $s_c$, and the latter represents the allowable maximum number of the $i$th token color in the place $s_c$; $n_c$ is the number of token colors used to express consumable resources.

### 3.2.3. Expression of Reusable Resources

Reusable resources are expressed by reusable-resource places. In order to reduce the complexity of the maintenance model, token colors are also used to express different kinds of reusable resources. Thus, there's also only one reusable-resource place in the model.

Because maintenance tasks utilizing reusable resources will also release the same reusable resources, transitions expressing these maintenance tasks will belong to both the pre-set and the post-set of the reusable-resource place. We can see that the pre-set and the post-set of reusable-resource places have common elements, and they are totally identical. During maintenance tasks utilizing reusable resources, the reusable resources will be occupied; and after these maintenance tasks have been finished, the reusable resources will be released. Therefore, the number of token colors in the reusable-resource places will not vary after transitions expressing these maintenance tasks have been fired.

Let $S_R = \{s_r\}$ be the set of reusable-resource places, their marking and capacity function can be expressed as

$$M(s_r) = m_{r1}c_{r1} + m_{r2}c_{r2} + \cdots + m_{rn_r}c_{rn_r}$$
$$K(s_r) = k_{r1}c_{r1} + k_{r2}c_{r2} + \cdots + k_{rn_r}c_{rn_r} \tag{7}$$

where $c_{ri}$ $(i = 1, 2, \cdots, n_r)$ is the $i$th token color expressing the $i$th type of reusable resources; $m_{ri}$ and $k_{ri}$ are both nonnegative integers, the former represents the number of the $i$th token color in the place $s_r$, and the latter represents the allowable maximum number of the $i$th token color in the place $s_r$; $n_r$ is the number of token colors used to express consumable resources. Places are drawn as circles in this study.

### 3.3. Relationship Expression for Maintenance Tasks, States and Resources

In Petri nets, arcs establish the link between transitions and places. In the maintenance process, they are used to denote the relationship between maintenance tasks and system states, or between maintenance tasks and resources. To arcs originating from a place to a transition, their weight functions denote the occurrence condition of the maintenance tasks expressed by the transition. To arcs originating from a transition to a place, their weight functions denote the changes of system states or maintenance resources caused by the maintenance task expressed by the transition.

To arcs connecting with transitions and state places, i.e., for all $f \in S_S \times T \cup T \times S_S$, their weight function can be expressed as:

$$W(f) = c_s \tag{8}$$

To arcs connecting with transitions and consumable-resource places, i.e., for all $f \in S_C \times T \cup T \times S_C$, their weight function can be expressed as:

$$W(f) = w_{c1}c_{c1} + w_{c2}c_{c2} + \cdots + w_{cn_c}c_{cn_c} \tag{9}$$

where $w_{ci}$ $(i = 1, 2, \cdots, n_c)$ is a nonnegative integer, and we have $w_{c1} + w_{c2} + \cdots w_{cn_c} \neq 0$.

To arcs connecting with transitions and reusable-resource places, i.e., for all $f \in S_R \times T \cup T \times S_R$, their weight function can be expressed as:

$$W(f) = w_{r1}c_{r1} + w_{r2}c_{r2} + \cdots + w_{rn}c_{rn} \tag{10}$$

where $w_{ci}$ $(i = 1, 2, \cdots, n_c)$ is also a nonnegative integer, and we also have $w_{c1} + w_{c2} + \cdots w_{cn_c} \neq 0$.

### 3.4. Structure Expression of Maintenance Process

The structures of the maintenance processes reflect the logical relationships among maintenance tasks. In a maintenance process, there may be three kinds of structures, which are serial, parallel and competitive.

#### 3.4.1. Expression of Serial Structures

If several maintenance tasks in a maintenance process are carried out one by one in sequence, they will have a serial relationship and the maintenance process will have a serial structure. To a status place in the Petri net, the element of its pre-set and the element of its post-set have a serial relationship. Namely, for all $s_s \in S_S$, if $t_i \in \bullet s_s$ and $t_j \in s_s \bullet$, then $t_i$ and $t_j$ have a serial relationship. A typical serial structure is given in Figure 2.

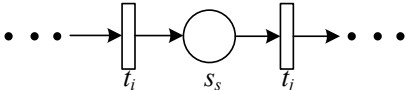

**Figure 2.** The serial structure.

In Figure 2, $t_i$ and $t_j$ have a serial relationship.

#### 3.4.2. Expression of Parallel Structures

If several maintenance tasks in a maintenance process are carried out simultaneously, they will have a parallel relationship and the maintenance process will have a parallel structure.

To different transitions in the Petri net, if elements of their pre-sets (post-sets) belong to the post-set (pre-set) of the same transition, these transitions have a parallel relationship. Namely, for $s_{si} \in S_S$ and $s_{sj} \in S_S$, if $s_{si} \in \bullet t_i(t_i\bullet)$, $s_{sj} \in \bullet t_j(t_j\bullet)$ and $s_{si} \in t\bullet(\bullet t)$, $s_{sj} \in t\bullet(\bullet t)$, then $t_i$ and $t_j$ have a parallel relationship. A typical parallel structure is given in Figure 3.

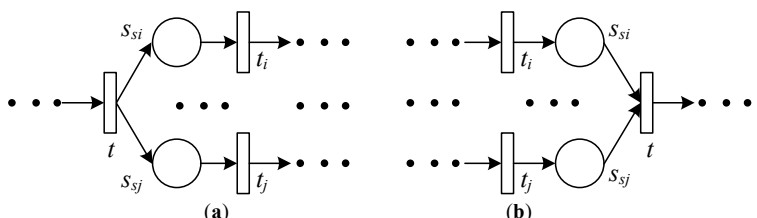

**Figure 3.** The parallel structure. (**a**) several maintenance tasks start simultaneously; (**b**) several maintenance tasks end simultaneously.

In both Figure 3a,b, $t_i$ and $t_j$ have a parallel relationship.

#### 3.4.3. Expression of Competitive Structures

Among several maintenance tasks belonging to a maintenance process, if only one of them can be carried out, the several maintenance tasks will have a competitive relationship and the maintenance process will have a competitive structure. In a Petri net, different transitions belonging to the same

post-set of a status place have a competitive relationship. Among all these transitions, only one of them can be fired. A typical competitive structure is given in Figure 4.

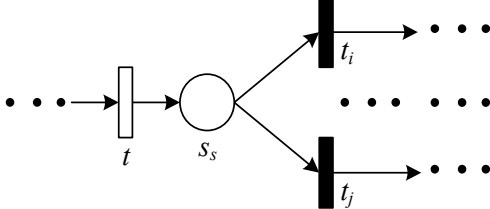

**Figure 4.** The competitive structure.

In Figure 4, $t_i$ and $t_j$ are both immediate transitions, and they have a competitive relationship, namely, only one of them can be fired. In maintenance processes, the competitive structure is usually used to express relationships among maintenance tasks after fault diagnosis. Take Figure 4 as an example, the transition $t$ can express the task of fault diagnosis. When there is a token in the state place $s_s$, it means the task of fault diagnosis has been completed, and which unit should be replaced or restored has been decided. Then, transition $t_i$ and transition $t_j$ will be used to express the replacing or restoring task of unit $i$ and unit $j$. For all $t_i \in s_s \bullet$ $(i = 1, 2 \cdots n)$, the probability that $t_i$ can be fired is expressed as

$$p_i = \frac{\lambda_i}{\sum\limits_{i=1}^{n} \lambda_i} \tag{11}$$

where $\lambda_i$ is the failure rate of the $i$th unit pertinent to transition $t_i$, $n$ is the number of transitions in the post-set of $s_s$.

## 4. Maintainability Evaluation Approaches Using the Maintenance Process Model

By using the maintenance process model based on SCPN, methods of calculating maintenance time and resources are proposed in this section, then the system maintainability can be evaluated.

### 4.1. Maintenance Resource Calculation

To determine the number of required resources, we first have to decide which maintenance tasks have been conducted. In other words, we have to decide which transitions have been fired. The decision can be made via the state function of the SCPN, which is expressed as

$$M_U = M_0 + C \times U \tag{12}$$

where $M_0$ is the initial marking, $M_U$ is the ultimate marking, $U$ is a vector whose element $u_i$ denotes the firing number of the transition $t_i$, $C$ is the incidence matrix and its element can be express as

$$c_{ij} = W(t_j, s_i) - W(s_i, t_j) \tag{13}$$

As a maintenance task can be conducted only once, $u_i$ can take the value either 0 or 1. We use $T_F$ to denote the set of the fired transitions, if $u_i = 1$, then we have $t_i \in T_F$.

#### 4.1.1. Method of Calculating Consumable Resources

To arcs originating from consumable-resource places to transitions ($f \in S_C \times T$), the coefficients of their weight functions indicate the number of consumable resources required for conducting maintenance tasks expressed by the corresponding transitions. Take Equation (9) as an example, we can determine that the required number of the $i$th consumable resource is $w_{ci}$. As consumable resources will be consumed in the relevant maintenance tasks, the total required consumable resources in the

maintenance process will be the summation of all consumable resources consumed in the maintenance tasks. Hence the required number of the *i*th consumable resource can be expressed as

$$N_{ci} = \sum_{f \in S_C \times T_F} w_{ci}(f) \qquad (14)$$

where $w_{ci}$ is the *i*th coefficient in the weight function of the arc *f*.

### 4.1.2. Method of Calculating Reusable Resources

To arcs originating from reusable-resource places to transitions ($f \in S_R \times T$), the coefficients of their weight functions indicate the required number of reusable resources. Take Equation (10) as an example, we can determine that the required number of the *i*th reusable resource is $w_{ri}$. As reusable resources will not be consumed in each maintenance task of the maintenance process, the total required reusable resources in the process will be the maximum number of the reusable resources consumed in each maintenance task. Hence the required number of the *i*th reusable resource can be expressed as

$$N_{ri} = \max_{f \in S_R \times T_F} w_{ri}(f) \qquad (15)$$

where $w_{ri}$ is the *i*th coefficient in the weight function of the arc *f*.

### 4.2. Maintenance Time Simulation

When all the firing rates of transitions are constant, the SCPN will be isomorphic to a continuous-time Markov chain. However, time durations of maintenance tasks may follow various distributions, the continuous-time Markov chain cannot be applied to obtain the maintenance time in this situation. We will use a discrete-event simulation to obtain the maintenance time.

The input of our discrete-event simulation procedure includes: (1) the input incidence matrix $W^- = [W(s_i, t_j)]$; (2) the output incidence matrix $W^+ = [W(t_j, s_i)]$; (3) the initial marking $M_0$; (4) the capacity function $K(s_s)$, $K(s_c)$ and $K(s_r)$; (5) the firing time distribution of each timed transition; (6) the probability that the immediate transition $t_i$ can be fired, namely $p_i$ given in Equation (11).

The flowchart of the discrete-event simulation procedure is given in Figure 5.

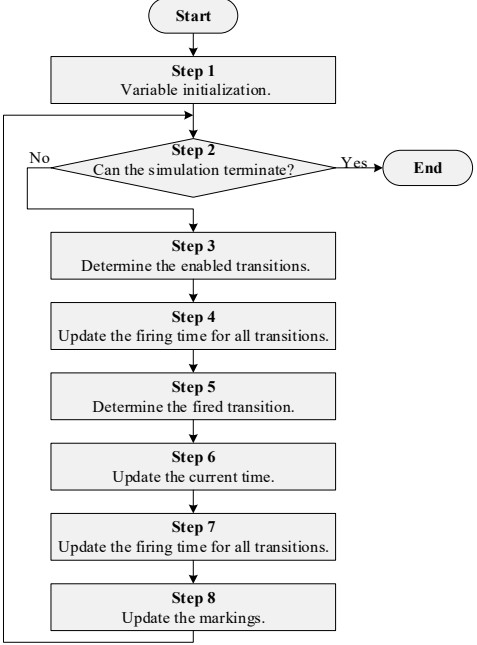

**Figure 5.** The discrete-event simulation for maintenance time.

We can get one sample of the maintenance time from one simulation, and we can get the mean maintenance time of the maintenance process from several simulations.

The detailed step of one discrete-event simulation is as follows:

Step 1: initialize variables

Let $M = M_0$, $\pi_{current} = 0$, and $\pi_j = 0$ for all $j$; $\pi_{current}$ is the current time, $\pi_j$ is the firing time of the transition $t_j$, and $M$ is the current system marking.

Step 2: decide whether the simulation can terminate

The simulation will terminate when $M(s_s^{(u)}) = c_s$, $s_s^{(u)}$ is the state place that denotes the ultimate maintenance state. In this situation, $\pi_{current}$ will be a sample of the maintenance time. If $M(s_s^{(u)}) \neq c_s$, the simulation will go to Step 3.

Step 3: determine the enabled transitions

A Boolean variable $E_j$ is used to denote whether the transition $t_j$ is enabled or not. When $E_j$ equals 1, $t_j$ is enabled; otherwise, $t_j$ is not enabled. We let $E_j = 0$ for all $j$. We can determine which transitions are enabled by Equation (3). If $t_j$ is enabled, let $E_j = 1$.

Step 4: update the firing time for all transitions

For each non-enabled transition ($E_j = 0$), let its firing time equal 0 ($\pi_j = 0$). For each enabled transition ($E_j = 1$), if its original firing time is 0 ($\pi_j = 0$), a random variable will be generated as its new firing time according to its firing time distribution; otherwise, its firing time will not be changed.

Step 5: determine the fired transition

Among all the enabled transitions, the transition that has the minimum firing time ($\pi_{min}$) will be fired. If $\pi_{min} = 0$, it means an immediate transition will be fired. In this situation, there might be several enabled immediate transitions, and the fired transition will be selected in terms of Equation (11).

A Boolean variable $F_j$ is used to denote whether the transition $t_j$ is enabled or not. If $F_j$ equals 1, $t_j$ is enabled; otherwise, $t_j$ is not enabled.

Step 6: update the current time

The current time $\pi_{current}$ will be updated by $\pi_{current} + \pi_{min}$, i.e., let $\pi_{current} = \pi_{current} + \pi_{min}$.

Step 7: update the firing time for all enabled transitions

The firing time of the enabled transition $t_j$ ($\pi_j$) will be updated by $\pi_j - \pi_{min}$, i.e., let $\pi_j = \pi_j - \pi_{min}$.

Step 8: update the markings

The markings will be updated via the state function of the Petri net, i.e., $M$ will be updated by $M + C \times F$ ($M = M + C \times F$). The $j$th element of $F$ is the Boolean variable $F_j$. Finally, go back to Step 2.

## 5. Case Study

In the section, the line maintenance of a wheel steering system installed on a regional jet's nose landing gear is used to illustrate the application of our proposed methods. The wheel steering system is given in Figure 6.

The steering control motor, powered by the hydraulic source, is installed on the shock strut. The upside of the toque link is connected with the steering control motor, and its downside is connected with the wheel axle installed on the sliding tube. Hence the steering motor can turn the nose wheel via the sliding tube and torque link around the shock strut axle.

We make the assumption that the system includes two line replaceable units, the steering control motor and the toque link, and their failure rates are $1.25 \times 10^{-4}$ per hour and $3.12 \times 10^{-5}$ per hour respectively. The line maintenance process of the wheel steering system includes two maintenance events, which are the replacement of the steering control motor and the replacement of the toque link. We assume that commonly used resources have already been available before the maintenance process, only the spare parts of the steering control motor are not available.

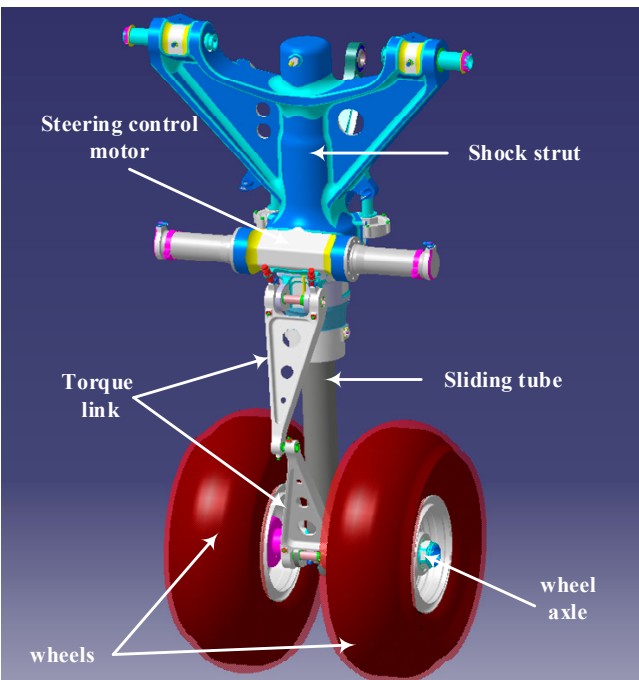

**Figure 6.** The structure of the wheel steering system.

The maintenance process of the wheel steering system based on the SCPN is shown in Figure 7.

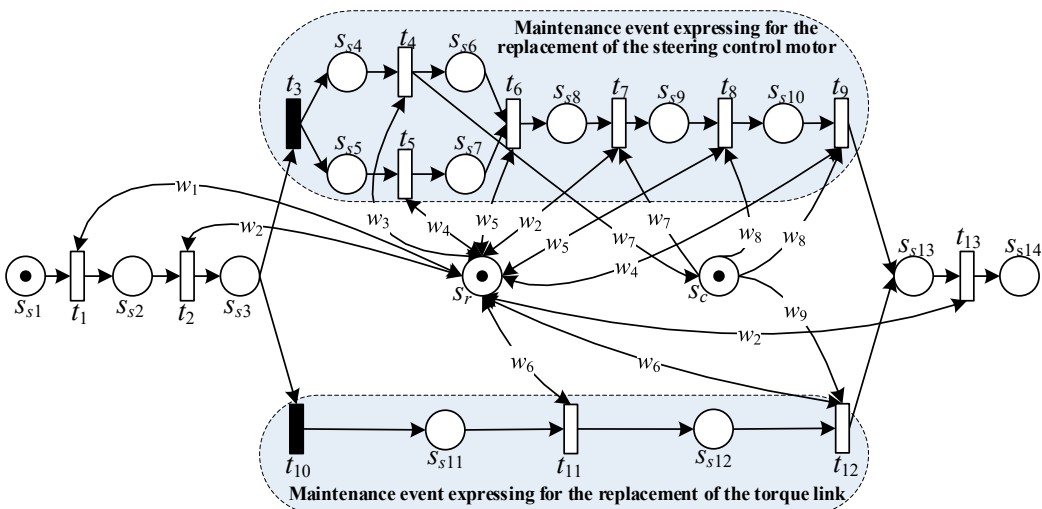

**Figure 7.** Maintenance process of the wheel steering system.

In Figure 7, $s_{s1}$ to $s_{s14}$ are state places, they only have one kind of token color $c_s$. $s_c$ is the consumable-resource place, and it has four kinds of token colors $c_{c1}$, $c_{c2}$, $c_{c3}$ and $c_{c4}$, which are used to denote spare parts of steering control motors, spare parts of toque links, washers and cotter pins, respectively. $s_r$ is the reusable-resource places, and it has six token colors $c_{r1}$, $c_{r2}$, $c_{r3}$, $c_{r4}$, $c_{r5}$ and $c_{r6}$, which are used to denote maintenance staffs, spanners, pliers, screwdrivers, lifting jacks and diagnostic equipment. $t_1$ to $t_{13}$ are transitions, among which, $t_3$ and $t_{10}$ are immediate transitions and others are timed transitions.

Table 1 lists the maintenance tasks expressing by transitions in Figure 6 as well as the pertinent maintenance time distributions. The maintenance time distributions are obtained by statistical analysis of maintenance time samples collected from the line maintenance of a certain type of civil aircraft.

**Table 1.** Detailed information of transitions.

| Transition | Maintenance Task | Distribution | Parameter (Minutes) |
|:---:|:---:|:---:|:---:|
| $t_1$ | Hoist the aircraft | Lognormal | $\mu = 1.28, \sigma = 1.16$ |
| $t_2$ | Fault diagnosis | Lognormal | $\mu = 0.88, \sigma = 1.14$ |
| $t_3$ | Not applicable | Not applicable | Not applicable |
| $t_4$ | Get the spare part of the steering control motor | Exponential | $\theta = 5.53$ |
| $t_5$ | Dismantle hoses and cables | Lognormal | $\mu = 1.12, \sigma = 0.29$ |
| $t_6$ | Dismantle connectors (Nuts and screws) | Normal | $\mu = 0.95, \sigma = 0.06$ |
| $t_7$ | Replace the steering control motor | Exponential | $\theta = 1.2$ |
| $t_8$ | Install the connectors (Nuts and screws) | Normal | $\mu = 1.35, \sigma = 0.06$ |
| $t_9$ | Install hoses and cables | Lognormal | $\mu = 1.43, \sigma = 0.24$ |
| $t_{10}$ | Not applicable | Not applicable | Not applicable |
| $t_{11}$ | Dismantle the torque link | Normal | $\mu = 0.9, \sigma = 0.08$ |
| $t_{12}$ | Install the new torque link | Normal | $\mu = 1.22, \sigma = 0.06$ |
| $t_{13}$ | Adjustment | Lognormal | $\mu = 0.91, \sigma = 0.29$ |

For all $f \in S_S \times T \cup T \times S_S$, the weight function can be expressed as $W(f) = c_s$. For other arcs, their weight functions $w_1 \sim w_9$ are expressed as $2c_{r1} + c_{r5}$, $2c_{r1} + c_{r5} + c_{r6}$, $c_{r1} + c_{r5}$, $2c_{r1} + 2c_{r2} + 2c_{r3} + c_{r5}$, $2c_{r1} + 2c_{r2} + 2c_{r3} + 2c_{r4} + c_{r5}$, $c_{r1} + c_{r2} + c_{r3} + c_{r4} + c_{r5}$, $c_{c1}$, $4c_{c3} + 2c_{c4}$ and $c_{c2} + 4c_{c3} + 2c_{c4}$, respectively. By Equation (13), the incidence matrix C can be calculated as

$$
C_{s \times t} = 
\begin{bmatrix}
-c_s & 0 & 0 & 0 & 0 & 0 & 0 & 0 & 0 & 0 & 0 & 0 & 0 \\
c_s & -c_s & 0 & 0 & 0 & 0 & 0 & 0 & 0 & 0 & 0 & 0 & 0 \\
0 & c_s & -c_s & 0 & 0 & 0 & 0 & 0 & 0 & -c_s & 0 & 0 & 0 \\
0 & 0 & c_s & -c_s & 0 & 0 & 0 & 0 & 0 & 0 & 0 & 0 & 0 \\
0 & 0 & c_s & 0 & -c_s & 0 & 0 & 0 & 0 & 0 & 0 & 0 & 0 \\
0 & 0 & 0 & c_s & 0 & -c_s & 0 & 0 & 0 & 0 & 0 & 0 & 0 \\
0 & 0 & 0 & 0 & c_s & -c_s & 0 & 0 & 0 & 0 & 0 & 0 & 0 \\
0 & 0 & 0 & 0 & 0 & c_s & -c_s & 0 & 0 & 0 & 0 & 0 & 0 \\
0 & 0 & 0 & 0 & 0 & 0 & c_s & -c_s & 0 & 0 & 0 & 0 & 0 \\
0 & 0 & 0 & 0 & 0 & 0 & 0 & c_s & -c_s & 0 & 0 & 0 & 0 \\
0 & 0 & 0 & 0 & 0 & 0 & 0 & 0 & 0 & c_s & -c_s & 0 & 0 \\
0 & 0 & 0 & 0 & 0 & 0 & 0 & 0 & 0 & 0 & c_s & -c_s & 0 \\
0 & 0 & 0 & 0 & 0 & 0 & 0 & 0 & c_s & 0 & 0 & c_s & -c_s \\
0 & 0 & 0 & 0 & 0 & 0 & 0 & 0 & 0 & 0 & 0 & 0 & c_s \\
0 & 0 & 0 & 0 & 0 & 0 & 0 & 0 & 0 & 0 & 0 & 0 & 0 \\
0 & 0 & 0 & w_7 & 0 & 0 & -w_7 & -w_8 & -w_8 & 0 & 0 & -w_9 & 0
\end{bmatrix}
\tag{16}
$$

The initial markings of the maintenance process based on SCPN can be expressed as

$$
\begin{aligned}
M_0(s_{si}) &= \begin{cases} c_s & i = 1 \\ 0 & i \neq 1 \end{cases} \\
M_0(s_c) &= m_{0c1}c_{c1} + m_{0c2}c_{c2} + m_{0c3}c_{c3} + m_{0c4}c_{c4} \\
M_0(s_r) &= m_{0r1}c_{r1} + m_{0r2}c_{r2} + m_{0r3}c_{r3} + m_{0r4}c_{r4} + m_{0r5}c_{r5} + m_{0r6}c_{r6}
\end{aligned}
\tag{17}
$$

where $m_{0c1} = 0$, the reason is that the spare parts of the steering control motor are unavailable; all other coefficients are large integers, which means the related resources are enough for the requirements of the corresponding maintenance tasks.

The ultimate marking of the maintenance process based on SCPN can be expressed as

$$
\begin{aligned}
M_u(s_{si}) &= \begin{cases} c_s & i = 14 \\ 0 & i \neq 14 \end{cases} \\
M_u(s_c) &= m_{uc1}c_{c1} + m_{uc2}c_{c2} + m_{uc3}c_{c3} + m_{uc4}c_{c4} \\
M_u(s_r) &= m_{ur1}c_{r1} + m_{ur2}c_{r2} + m_{ur3}c_{r3} + m_{ur4}c_{r4} + m_{ur5}c_{r5} + m_{ur6}c_{r6}
\end{aligned}
\tag{18}
$$

where the coefficients $m_{uc1} \sim m_{uc4}$ and $m_{ur1} \sim m_{ur6}$ are unknown, because we do not know what resources may be left after the maintenance process. According to Equations (12) and (16)–(18), we can get two solutions of $T_F$

$$
\begin{aligned}
T_{F1} &= \{t_1, t_2, t_3, t_4, t_5, t_6, t_7, t_8, t_9, t_{13}\} \\
T_{F2} &= \{t_1, t_2, t_{10}, t_{11}, t_{12}, t_{13}\}
\end{aligned}
\tag{19}
$$

When transitions in $T_{F1}$ are fired, it means the maintenance event expressing the replacement of the steering control motor is conducted. According to Equations (14) and (15), we can get the required quantities of resources, which are one steering control motor, eight washers, four cotter pins, two maintenance staffs, two spanners, two pliers, two screwdrivers, one lifting jack and one diagnostic equipment. When transitions in $T_{F2}$ are fired, it means the maintenance event expressing the replacement of the torque link is conducted. In the same way, we determine that one toque link, four washers, two cotter pins, two maintenance staffs, one spanner, one plier, one screwdriver, one lifting jack and one diagnostic equipment are required for this maintenance event.

We get 1000 maintenance time samples in terms of the discrete-event simulation proposed in Section 4.2. According to these samples, the mean of the maintenance time is 22.28 min, the standard deviation of the maintenance time is 7.56 min and 89.2% of the maintenance time samples are less than half an hour, which is the up limit of the line maintenance for civil aircraft. The elapsed time of our discrete-event simulation procedure running in MATLAB is less than one second. The computer used was Intel (TM) i5-6500 four Core Processor 3.2 GHz, 8 GB of RAM.

We have also developed a GRASP based method [1] to obtain the maintenance time samples. In terms of these samples, the mean of the maintenance time is 21.6 min, the standard deviation of the maintenance time is 7.39 min and 90.5% of the maintenance time samples are less than half an hour. It is proved that the results of our method are very close to the results of the GRASP based method.

The histograms of the maintenance time samples obtained by the two methods are given in Figure 8. Figure 8 shows the maintenance time follows a bimodal distribution, as the maintenance process is composed of two maintenance events, one is the replacement of the toque link, and the other is the replacement of steering control motor.

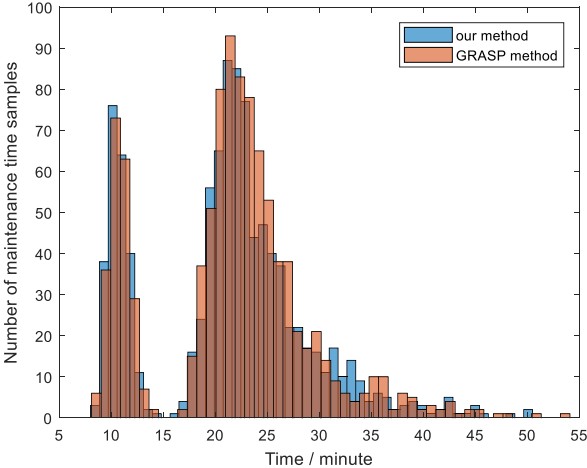

**Figure 8.** The histogram of maintenance time samples.

## 6. Conclusions

In the paper, a maintenance process model is developed by using the SCPN. The tuples of the SCPN, including places, tokens, transitions and arcs, are used to express the constituents of the maintenance process, such as maintenance states, maintenance resources, maintenance tasks, and their relationships. Then, different kinds of maintenance process structures are expressed by the SCPN. Based on our proposed maintenance process model, methods for calculating maintenance resources are given, and a discrete-event simulation procedure is proposed for maintenance time evaluation. Our approaches have the following advantages:

(1) Compared with the traditional maintainability evaluation method given in MIL-STD-471A, the maintenance demonstration is not required to be conducted for the whole maintenance process, maintenance time samples of the whole maintenance process can be obtained by our discrete-event simulation procedure in terms of the time distribution of each maintenance task.

(2) As the constituents of the maintenance process including maintenance tasks, system states, maintenance resources and their relationships are expressed by tuples of the SCPN in our maintenance process model, the detailed information related to maintenance resources of each maintenance tasks can be obtained. Whereas, the requirements of maintenance resources are neglected in most maintainability evaluation methods, particularly the processed based maintainability evaluation method.

(3) As the maintenance time and the requirements of maintenance personnel, tools or equipment, and spare parts can be calculated via our proposed methods, both the time related parameters and the economic parameters of the system maintainability can be evaluated quantitatively. Whereas, the multiple attribute decision based maintainability evaluation method can only present a general and qualitative assessment of system maintainability.

**Author Contributions:** Conceptualization, Z.L.; methodology, Z.L. and J.L.; software, Z.L. and X.L.; formal analysis, Z.L., J.L. and X.L.; data curation, J.L. and L.D.; writing—original draft preparation, Z.L. and L.D.; project administration, Z.L.; funding acquisition, Z.L.

**Funding:** This research was funded by the National Natural Science Foundation of China with the grant number U1733124, and it is also supported by the Fundamental Research Funds for the Central Universities of NUAA with the grant number 3082018NT2018019.

**Conflicts of Interest:** The authors declare no conflict of interest.

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
