# Peer review of "Maintenance Process Simulation Based Maintainability Evaluation by Using Stochastic Colored Petri Net"

_applsci, doi:10.3390/app9163262_

Round 1
Reviewer 1 Report
The paper models a maintenance process with colored stochastic petri nets.
It is unclear to me, what a reader should learn from the paper.
a certain process can be modeled with CSPN? Yes, that is known.
Pages three to nine deal with defining CSPN. This is superfluous, the syntax and semantics can be read in literature.
In the maintenance example, there are a lot of assumptions about process times and the process itself. It is not demonstrated or validated that the assumptions have a connection to a certain real process.
Consequently, the conclusions are short.
Author Response
see the attached cover letter

Reviewer 2 Report
The paper deals with an up-to-date subject and is original, as far as I know.
Just one general question: why you used SCPN instead of Petri nets? Or why not to use Extended SCPN? One short explanation should be given in the text to explain your choice.
Some small spelling mistakes:
In the Introduction paragraph here too much "and"'s. For instance, on line 25 Maintainability, a characteristic of design and affected ... in my opinion the and should disappear.
Page 9, Figure 5, Step 2 - please correct the phrase. Maybe you can use: Can the simulation terminate?
Page 10, line 364 - It can not be CxF is not always 1. If is that so M is not equal to M+CxF. Is that so?
Pages 10 and 11 - where is toque link, should be torque link.
Reviewer 3 Report
The paper presents a maintenance process model based on Stochastic Colored Petri Nets with the aim of calculating the necessary maintenance time and resources for production processes.
The topic is worth of interest, nevertheless the article should be improved and made more consistent under various aspects.
First and foremost, the contribution and novelty of the paper has to be presented more clearly and evidently. At the moment the paper presents an application of colored petri nets for the calculation of performance indicators that can be useful for the maintainability evaluation. Nevertheless, such kind of application is not particularly novel in the context of systems evaluation with Petri nets. I suggest authors to better position the article and make it more solid with respect to the existing literature.
The title of the article can be made less convoluted and more direct, e.g., Evaluation of process maintainability by using Stochastic Colored Petri Nets.
The introduction presents a long list of articles that have tackled the maintenance evaluation problem with various techniques. Such discussion has to be improved and better organized. First, a comparison of the various contributions must be taken into account, otherwise this part of the section provides a low contribution in positioning the paper with respect to the state of the art in the related context. Moreover, no discussion is provided on the use of Petri nets for the modelling and simulation of production processes. Petri nets are a very powerful mathematical and graphical formalism when evaluating the performance of production process and authors have first to provide a wide discussion on this aspect and consequently motivate consistently the choice of Colored Petri nets. The discussion should include, but obviously it should not be limited to, the following papers:
DOI: 10.1109/RAM.2017.7889767
DOI: 10.1109/JSEE.2015.00071
DOI: 10.1002/sys.21462
DOI: 10.1016/j.jclepro.2015.09.100
DOI: 10.11591/eei.v7i1.845
DOI: 10.1080/15732479.2016.1190767
DOI: 10.1016/j.ifacol.2018.06.311
DOI: 10.1177/0954409715619453
The maintainalbility process evaluation is presented as an offline process. Nevertheless, the description of computation time has to be provided as well as information regarding the simulation and modelling environment.
